# Induction of Earlier Flowering in Cassava through Extended Photoperiod

**Marcela Pineda** [1,2]**, Nelson Morante** [1]**, Sandra Salazar** [1] **, Juan Cuásquer** [1]**, Peter T. Hyde** [3]**, Tim L. Setter** [3] **and Hernán Ceballos** [1,*]

[1]   The Alliance of Bioversity International and the International Center for Tropical Agriculture (CIAT), Apartado Aéreo 6713, Cali 763537, Colombia; lmpinedav@unal.edu.co (M.P.); n.morante@cgiar.org (N.M.); S.M.Salazar@cgiar.org (S.S.); j.b.cuasquer@cgiar.org (J.C.)

[2]   Universidad Nacional de Colombia, Sede Palmira, Valle del Cauca 763537, Colombia

[3]   Section of Soil and Crop Sciences, School of Integrative Plant Science, Cornell University, Ithaca, NY 14850, USA; pth7@cornell.edu (P.T.H.); tls1@cornell.edu (T.L.S.)

*   Correspondence: h.ceballos@cgiar.org or HernanCeballosL54@gmail.com; Tel.: +57-316-645-1508

**Abstract:** Erect plant architecture is preferred by farmers but results in late and scarce flowering, which slows down breeding considerably. Inducing earlier and abundant flowering in crossing nurseries (involving erect genotypes) is a key objective for cassava and was the subject of this study. Five genotypes with contrasting flowering behavior were grown under dark night (DN) and extended photoperiod (EP) conditions for three seasons. EP was achieved with different red light emitting diodes (LEDs) with 625–635 nm wavelength all night long or through night-breaks. EP reduced height and number of days to first branching, particularly in non- or late-flowering genotypes. A minimum of 0.02 μmol m$^{-2}$ s$^{-1}$ was required to elicit earlier flowering in plants illuminated all night. Early results using five genotypes were validated across 116 genotypes planted in a crossing nursery. EP promoted earlier flowering in erect-plant genotypes but reduced the number of branching events in early flowering genotypes to some extent. 50W LED lamps, fixed at 3 m above ground in a 4.5 m grid, proved to be a practical approach to extend photoperiod in breeding nurseries. Night breaks also proved effective, thus opening the possibility of using solar panels where electricity is not available.

**Keywords:** genetic gains; genetic recombination; genomic selection; inbreeding; plant architecture; speed breeding

## 1. Introduction

Cassava (*Manihot escultenta* Crantz) is a crop of Neotropical origin and has significant economic relevance, particularly in the lowland tropics. Its main product is the starchy roots that are harvested, usually, about 12 months after planting. It is a fundamental food security crop in many regions of the world, particularly Sub-Saharan Africa. It is also an important industrial crop (the second most important source of starch worldwide) [1]. Cassava is a perennial species usually grown as an annual crop. The roots can be harvested from six to 24 months after planting (MAP). More typically, however, farmers harvest cassava 10–12 MAP, at the end of the dry season (or before the cold season in subtropical regions), when root quality would be optimum, and store the stems only for a few weeks until the arrival of the rains (or the warm season in subtropical environments).

Cassava can be propagated either vegetatively from stems or sexually from seeds, but the former is the most common practice for the commercial production. Farmers commonly grow clonal hybrids. Before harvesting the roots the main stems (0.5–2.0 m in length, depending on the cultivar and growing

conditions) are cut and tied together in bundles of approximately 50 stems [2]. Farmers frequently prefer erect, non-branching types because this plant architecture facilitates cultural practices including mechanization, results in good production of vegetative planting material, and its transport and storage is easier. In addition, longer stems can withstand a longer storage period and may be a key trait to deal with the erratic rain patterns and warmer temperatures associated with climate change [3].

Sexual reproduction, the key requirement for crop breeding, is common and relatively easy to achieve in cassava [4,5]. Cassava is a diclinous and monoecious species producing either female (pistillate) or male (staminate) flowers in inflorescences (racemes or panicles) within the same plant. Female flowers are reduced to a pistil that is protected by petal-like bracts of leaf origin. What is commonly identified as male flowers are actually inflorescences of 10 single-stamen flowers. Bracts also protect these inflorescences, known as cyathia. Male and female cyathia will be treated in this article as if they were single flowers as the distinction is only relevant from the botanical point of view [6].

Male flowers are more numerous and develop in the upper section of the inflorescence [6–8]. The proportion of female flowers is considerably lower than that of male flowers. Female flowers are found in the proximal branches of the inflorescence and their anthesis occurs about 14 days earlier than that of male flowers (protogynia). Cassava naturally outcrosses and bees are the most common pollinators. Self-pollination can occur between male and female flowers from different branches or plants of the same genotype. Stigmas remain receptive for at least three days. In general, the process of fertilization of the egg cell and fruit/seed set is highly irregular and rather inefficient [6,9–12].

Inflorescences always develop at the apex of the developing stem. Sprouting of the buds below the inflorescence allows further growth of the plant. Therefore, the plant first flowers and then develops branches [7]. Every flowering event, therefore, results in branching. Some genotypes flower early and several times (starting two or three MAP, and up to six times) during a growth cycle, and others flower little and late or not at all. Time and frequency of flowering (and thus plant architecture) are under genetic control and also affected by environment. Molecular markers for height of first branching have been identified [13]. Early branching genotypes tend to produce progenies that flower early and branch low. The relationship between flowering and plant architecture results in a dilemma for the breeder because the production of botanical seed from erect genotypes is sparse, slow and, ultimately, more expensive. Perhaps one of the most important areas of research to accelerate and improve the impact of breeding, therefore, would be the development of a protocol for temporarily inducing flowering in cassava crossing nurseries.

Flowering in plants is a complex process involving the interaction of environmental, developmental, and genetic factors [14–18]. The influence of photoperiod in the induction of flowering was discovered more than a century ago [19]. Nknott demonstrated in 1934 the occurrence of a mobile signal that elicits flowering in spinach [20]. Studies have shown that it is actually the length of the continuous dark period, not the daylength as such which elicits the photoperiod response. Hence, a night break with a relatively brief period of light in the middle of the night is interpreted by the plant as a short night. The terms "florigen" and Flowering Locus T (FT) have been used to identify this photoperiod stimulus, initiated in the leaves, but then transmitted to the apical meristem [21–26]. FT expression in leaves and its movement to the apex, where it triggers flowering, appears to be universally conserved [27,28]. Environmental conditions such as temperature and photoperiod signal FT and elicit flowering in many different plant species [16,29–34].

In the case of cassava, early efforts to induce early flowering were made on the exogenous applications of growth regulators [35] or the use of locations with longer photoperiods and cooler temperatures [36,37]. It has been demonstrated that nutritional conditions affect flowering behavior in cassava; production of flowers and fruits was greatly enhanced in the unfertilized (NPK) treatment across four different genotypes [38]. Induction of flowering for plants growing in vitro, through addition to the growth media of gibberellins and cytokinin in the presence of auxin growth regulators, has also been reported [39]. In 2011 Adeyemo et al. provided [40] the first report on the molecular genetics of a cassava circadian-clock component (MeELF4). Moreover, the over expression of the

Arabidopsis FT gene in cassava, achieved through genetic modification, hastens flower induction, indicating that cassava has the molecular factors to respond to FT signals [41]. It has been reported that cassava has two FT homologs, and expression of one of these FT homologs, MeFT2, was photoperiod dependent [42]. Induction of earlier flowering in cassava, through grafting, was recently reported [15,43]. Hyde et al. [44] reported that the anti-ethylene growth regulator silver thiosulfate improved early inflorescence and flower development as well as flower longevity such that flower numbers were increased. Pruning young branches soon after flowering has been induced prevents the common abortion of first inflorescences [45]. The additional spraying of benzyladenine in pruned plants also promotes flower development and often results in the feminization of male flowers [45].

The present study was carried out to: (1) assess the effect of extended photoperiod through illumination with red light emitting diodes (LEDs) during the night on flowering in cassava; (2) define the minimum light intensity required to elicit earlier flowering; and (3) develop a practical method to manipulate photoperiod conditions that could be used in cassava crossing nurseries.

## 2. Materials and Methods

The main objective of this work was to compare flowering patterns in plants growing under dark night (DN) and extended photoperiod (EP) conditions. Different sources of light, light intensities, genotypes, seasons, and duration of the EP were used to make such contrast. The search for a reliable, affordable, and efficient method required a continuous evolution of the methods evaluated. There was a need to demonstrate the impact of EP on one hand, but also an urgent need to develop practical approaches that breeders can implement, on the other.

### 2.1. Location

Experiments were conducted through three consecutive growing seasons (2016/17, 2017/18, and 2018/19) at the Centro Internacional de Agricultura Tropical (CIAT) Experimental Station, in Palmira, Valle del Cauca, Colombia. This site is located less than four degrees north of the Equator. The duration of natural photoperiod is therefore about 12 h and quite uniform throughout the year. The altitude of this location is 965 m above sea level. Night temperatures, therefore, tend to be lower than at locations in the same latitude at sea level. Historic average maximum and minimum temperatures through the year are 30.1 ± 2.7 and 19.2 ± 1.2, respectively. Different fields were used each year. Previous experience in crossing nurseries indicated that spatial variability represented a negligible contribution to variation in flowering behavior in this experimental station.

### 2.2. Germplasm

Five genotypes were chosen because of their contrasting flowering patterns. Two of these genotypes were previously found to respond differently to grafting with an early-flowering understock [15]: SM 3348-29 branched and flowered but SM 3409-43 did not branch or flower after grafting. None of these genotypes ordinarily flower. Three other genotypes were included because of their relevance and/or known flowering behavior. GM 971-2 begins flowering at 3–4 MAP. CM 4919-1 is a late flowering genotype (8–9 MAP). GM 3893-65, or "*Asparagus*" cassava, has leaves without petiole and does not flower within a normal growing cycle. In the third year of research, 116 elite genotypes from the CIAT breeding program were grown in a crossing nursery EP and DN conditions. This served to validate findings from the five selected genotypes on a wider, more diverse pool of germplasm.

### 2.3. Light Sources

Extended photoperiod conditions were achieved through the illumination of plants during the night with red (peak around 625–635 nm) light emitting diodes (LEDs). The following sources were used: (a) five individual LEDs per plant (5LEDs, Figure 1A); (b) ten individual LEDs per plant (10LEDs, Figure 1B); (c) one 20-cm LED tape per plant containing about 30 LEDs (30LEDs, Figure 1C); (d) two 20-cm LED tapes per plant (60LEDs, Figure 1D); and (e) 50 w LED lamps with reflectors

(Figure 1E). For treatments (a–d), LEDs and LED tapes (unknown manufacturer) were purchased and custom assembled in an electric supplies store to meet the required specifications for outdoor use. They were suspended about 10–30 cm above the apex of each plant. These light sources were raised periodically as plants grew, maintaining a close distance above the apex. In treatment (e) the lamps were in a fixed position 3 m above ground (Figure 1E). Lights were turned on at sunset and turned off at sunrise. Illumination of plants, therefore, lasted all night, except for Experiment 3, which evaluated the usefulness of night breaks.

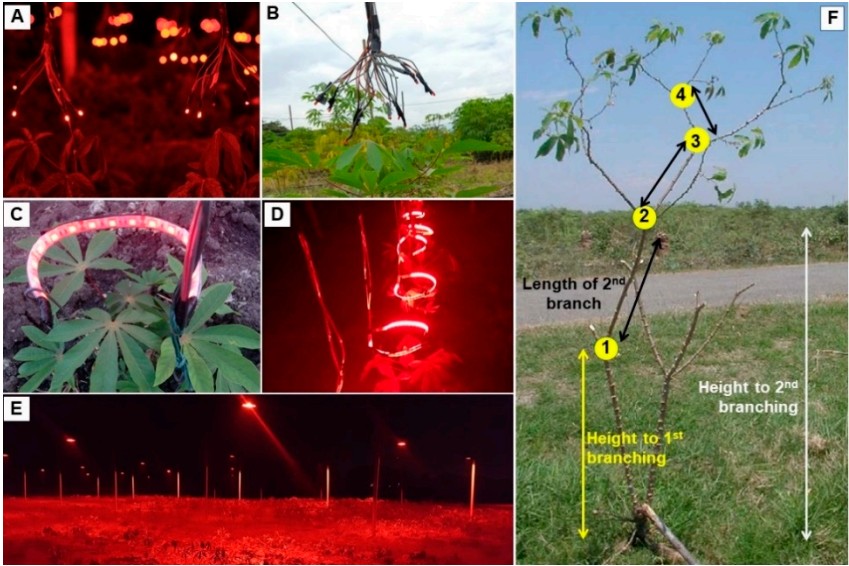

**Figure 1.** Illustration of the different sources of light emitting diode (LED) red lights used to extend the photoperiod in the different experiments reported in this study. (**A**): Five individual LEDs. (**B**): Ten individual LEDs. (**C**): One 20-cm LED tape. (**D**): Two 20-cm LED tapes. (**E**): The field illuminated with 50 W lamps with reflectors. (**F**): Illustration of a plant with four levels of branching (i.e., flowered four times during its growth). For height of first branch only one sprouted stem per plant was randomly considered as well as the branches above it.

### 2.4. Experiment 1: Photoperiod Extension with Individual LED Lamps above each Plant

In the first growing season (2016/17), the five genotypes were planted in the field on August 8, 2016 following a design similar to a split-plot design (imposed by restriction in the installation of the illumination system). There were six main plots defined by the illumination conditions at night: two main plots were not illuminated (DN) and served as checks. The four remaining main plots were illuminated at night: two with low-intensity light provided by individual LEDs (a–b above) and the remaining two, with high-intensity sources based on LED tapes (c–d above). Table 1 provides the light quantity provided by each of the four sources of light, measured with a SpectraPen mini (Photon Systems Instruments, Drásov, Czech Republic) at 10, 20 and 30 cm. Five measurements were taken per source of light and distance.

**Table 1.** Light quantity provided by different LED sources at 10, 20 and 30 cm (based on five measurements). Within parenthesis are the respective standard deviations. Illumination of plots began at different days after planting (DAP).

| Source of Light | Initiation of Illumination | Light Quantity ($\mu$mol/m$^2$/s) | | |
|---|---|---|---|---|
| | | 10 cm | 20 cm | 30 cm |
| **Individual light emitting diodes (LEDs)** | | | | |
| 5 LEDs | 11 DAP | 0.141 (0.003) | 0.060 (0.002) | 0.041 (0.001) |
| 10 LEDs | 23 DAP | 0.177 (0.011) | 0.070 (0.002) | 0.053 (0.001) |
| **LEDs Tapes** | | | | |
| 30 LEDs | 14 DAP | 0.685 (0.021) | 0.337 (0.003) | 0.186 (0.006) |
| 60 LEDs | 17 DAP | 1.513 (0.024) | 0.776 (0.008) | 0.388 (0.010) |

The five genotypes were randomly planted within each main plot in 5 m rows with ten plants per row. Rows were spaced 2 m apart. Plant density was the normal for cassava (10,000 plants ha$^{-1}$). Illumination was initiated as soon as the installation of the system was completed in each main plot (Table 1). The source of light was provided individually for each plant and maintained 10–30 cm above the apical shoot. As plants grew, the source of light had to be raised. Each plant was considered an experimental unit and thus, data were taken on individual plants in all the experiments in the present study. Data was registered on a daily basis after the first plants began to flower and branch through 210 days after planting (DAP). A plastic curtain separated each main plot and prevented illumination from neighboring main plots. The curtains were unrolled only during the night.

In the second year of Experiment 1, the same five genotypes were planted, in a different field on 2 June 2017 and grown under EP and DN. However, because of delays in installation of the electrical equipment, all plants were pruned back to the stake level on 14 July 2017 and allowed to regrow. The results from the previous season indicated that the 5LEDs treatment resulted in a significantly earlier flowering compared with the DN checks, but also that responses were often not as good compared with the three remaining EP treatments. Since the main objective for the second season was to validate the results from the first year, the only EP treatment used was the one with lowest light quantity (5LEDs). The rationale was that positive results for the 5LEDs treatment during the second season would also validate the impact of treatments with higher light quantity. Illumination (in EP plots) was maintained all night long and begun as soon as the plants were pruned back. Growing conditions and plant density were the same as in the 2016/17 growing season. Data was taken individually on each plant on a daily basis after the first plants began to flower and branch. Data was recorded through 210 DAP.

*2.5. Experiment 2: Photoperiod Extension with 50W LED Lamps Illuminating a Large Area*

The experience gained during the first year made evident that raising the light sources as the plants grew was labor intensive. In addition, the electrical set-up in field conditions demanded constant maintenance. During the 2017/18 growing season, a 50W LED lamp with parabolic reflector was tested as an alternative to the cumbersome individual-plant LEDs used in Experiment 1. A single 50W lamp was placed in a fixed position 3 m above ground at the center of the experimental plot. The experiment, planted on 18 July 2017, included ten rows with a spacing of 1.5 m. Each row was 13.5 m long and half of it was planted with 14 stem cuttings from GM 971-2 and the other half with 14 cuttings from CM 4919-1. Planting within the row was 50 cm apart. This set-up was designed to provide a gradient with a maximum light intensity immediately below the reflector and minimums farther away. In fact, it was expected that plants in the periphery of the plot received negligible illumination (below the threshold for activating the response to EP). There were 280 plants from the two genotypes, placed at different distances from the reflector. Each Distance × Genotype combination was replicated at least twice.

A hand held spectroradiometer for accurate analysis in the 325–1075 nm spectral range (ASD FieldSpec® HandHeld 2, Malvern Panalytical, Malvern, UK) was used to measure light intensity at different distances from the reflector.

During the second season evaluating the 50W LED lamp, four genotypes (GM 971-2, CM 4919-1, GM 3893-65 and SM 3348-29) were grown under EP and DN conditions. Planting took place on June 6, 2018 and illumination in the EP plots began immediately and lasted all night long. Single rows with 10 plants per genotype were used. Data was taken individually on each plant on a daily basis after the first plants began to flower and branch. Data was registered through 210 DAP. For the 2018/19 season several 50W lamps were installed in a square grid 4.5 m apart. The illuminated area of neighboring lamps overlapped considerably (Figure 1E), guaranteeing that every plant would receive enough stimulus. Although light intensity was not uniform around the experimental area, results from previous experiments indicated that every plant would receive a stimulus in excess of what was required to elicit earlier flowering. This set up was also used for Experiment 4 described below.

### 2.6. Experiment 3: Night Breaks

Six different night breaks (NBs) were evaluated to test whether all-night illumination was necessary. Only two contrasting genotypes (CM 4919-1 and GM 971-2) were used. The different NBs lasted 30, 60, 90, 120, 180 or 240 min and were timed to take place with a center at midnight (e.g., the 30-min NB, began at 11:45 PM and ended at 00:15 AM).

Different light sources (Figure 1, Table 1) were randomly allocated to the six NBs. The 60LEDs light source was used for the 90- and 240-min NBs; the 30LEDs light source was used for the 30-min NB; and the 10LEDs source was used for the 60-, 120-, and 180-min breaks. There was, therefore, a range of combinations between duration of NB and light intensity. NB began as soon as ten stem cuttings were planted in single rows (5 m long). Planting distance within the row was 50 cm. The '*asparagus*' genotype (GM 3893-65), was planted two weeks earlier at high density to create light barriers separating the different night breaks. This genotype has leaves without petioles, is very tall and does not branch or branches very late under EP. All of these characteristics make GM3893-65 an ideal live barrier. Data was taken individually on each plant on a daily basis after the first plants began flowering and branching. The two genotypes serve as biologically relevant replications of the combination of NB duration and light intensity. Data was recorded through 210 DAP.

### 2.7. Experiment 4: Validation of the Effect of Photoperiod Extension in a Crossing Nursery

The results obtained in the two previous seasons were based on a limited number of genotypes. This experiment aimed to test responses to EP (using 50W LED lamps installed 3 m above ground in a 4.5 m grid) in a wider set of germplasm. Photoperiod treatments (EP and DN) were given to 116 elite genotypes in the CIAT breeding crossing nursery. As in all previous experiments, each genotype × treatment combination was planted in a single row. However, in this case only eight plants per row were used. The same 116 genotypes were planted in an immediately adjacent area in the same field and grown under the same management, except that they were not illuminated at night. Data was taken for each plant 168 DAP.

### 2.8. Field Management

Field management followed the standard procedures for cassava at CIAT. A mixture of the pre-emergence herbicides Karmex and Dual Gold (Diuron and metolachlor, respectively) was applied 4–7 days before planting. Manual weeding was made as necessary. Crossing nurseries at CIAT follow a standard fertilization protocol. No fertilizers are applied to the soil. Foliar nutrient solutions are applied one and two months after planting (MAP) to promote vegetative development. At 4–5 MAP a different foliar fertilizer (rich in P and K) is applied to promote flowering. In this study, only the foliar fertilization to sustain vegetative growth was provided during the first two MAP (2 g L$^{-1}$ of Coljap-Arysta Lifescience, Colombia, S.A.: ammonium nitrogen 1%, urea nitrogen 29%,

$P_2O_5$ 7%, and $K_2O$ 6%). Irrigation was provided via surface/gravity distribution also as required. Pest pressure, particularly from whiteflies (*Aleurotrachelus socialis*), was monitored constantly and maintained under control.

### 2.9. Data Recorded

The experimental units were the individual plants within a 10-plant plot (8 plants in Experiment 4). Each combination of treatments was planted in a single row and data were taken on individual plants. This design was a result of restrictions imposed by the illumination system. Plots were visited daily and records were taken when a plant began flowering and branched. The following variables were considered: days to first flowering (D1F); height of first branching event (HGT); and number of branching levels (NBL) or flowering events. Only one sprouted stem per plant was considered when more than one bud from the planted cutting had sprouted (Figure 1F).

### 2.10. Statistical Analysis

Statistical analyses were conducted with SAS [46]. Regressions analyses were performed using the PROC REG procedure. The NOINT option was used for the regression analyses to force the regression line to pass at the origin on day 0. Analysis of variance was performed using the PROC GLM procedure. Tukey's Adjusted Multiple Comparisons test was used to assess differences among averages. Years of experience making crosses for the cassava breeding program provided a wealth of information regarding the flowering behavior of this crop which was relevant for the current research. Open-pollinated seed is obtained from poly-cross nurseries following a special planting design that favors panmixia [2]. In poly-cross nurseries the same genotype is planted in different parts of the field. Plants from the same genotype flower at the same time regardless of their position in the field. This unpublished experience was critical for the design and statistical analysis of the data.

Genotypes were used as replications to assess the impact of extended photoperiod. Rather than controlling environmental variation (as blocks do in a randomized complete block design) the replications in the present study address genetic variation, which is the critical factor to consider in the case of flowering.

## 3. Results

In general, the different trials grew well without relevant problems. Environmental conditions were representative for the location used. There was no major biotic problem.

### 3.1. Response to Extended Photoperiod Based on Different Sources of Red Light

Table 2 presents the analysis of variance (ANOVA) table for the first season in Experiment 1. The analysis considers three photoperiod treatments: normal photoperiod (DN), extended photoperiod with low light intensity based on 5LEDs and 10LEDs treatments (EP-L) and extended photoperiod with high light intensity based on 30LEDs and 60LEDs treatments (EP-H). The distinction between the two light intensities is arbitrary but simplifies the analysis. Each of these three light conditions were represented by two main plots with the same set of genotypes planted randomly within them. All main sources of variation (photoperiod, genotypes, and the Photoperiod*Genotype interaction) had significant effects ($p \leq 0.01$) on days D1F, HGT) and NBL.

**Table 2.** Analysis of variance for data from Experiment 1 (2016/17 season). Useful orthogonal contrasts were used to partition the sources of variation due to photoperiod and photoperiod * genotype interaction.

| Source of Variation | df [a] | Mean Squares | | |
|---|---|---|---|---|
| | | Days to 1st Flowering | Height 1st Branching | Branching Levels |
| | | (Number) | (cm) | (Number) |
| Photoperiod (P) | 2 | 93,645.0 ** | 91,588.7 ** | 30.1 ** |
| EP vs. DN | 1 | 183,085.6 ** | 148,509.3 ** | 60.2 ** |
| EP-L vs. EP-H | 1 | 4204.4 ** | 34,668.1 ** | 0.1 |
| Genotype (G) | 4 | 113,751.8 ** | 259,285.1 ** | 49.4 ** |
| P * G [b] | 8 | 7066.7 ** | 8675.8 ** | 3.7 ** |
| G1 (EP vs. DN) | 1 | 101,094.1 ** | 50,471.0 ** | 52.0 ** |
| G2 (EP vs. DN) | 1 | 1904.0 * | 298.2 | 0.7 * |
| G3 (EP vs. DN) | 1 | 16,708.8 ** | 58,476.7 ** | 4.4 ** |
| G4 (EP vs. DN) | 1 | 72,422.5 ** | 89,036.4 ** | 16.1 ** |
| G5 (EP vs. DN) | 1 | 38,736.1 ** | 13,573.3 ** | 10.2 ** |
| Error | 285 | 295.3 | 606.8 | 0.2 |
| Total | 299 | | | |

[a] Degrees of freedom for Error and Total sources of variation of days to 1st flowering were 281 and 295, respectively. [b] G1 = CM 4919-1; G2 = GM3893-65; G3 = GM 971-2; G4 = SM3348-29; and G5 = SM3409-43. *, ** Statistically significant at 0.05 and 0.01 probability level, respectively.

Two orthogonal contrasts were made using the two degrees of freedom from the photoperiod source of variation. One contrast compares the averages of DN versus EP (across EP-L and EP-H) and is the main subject of this research. EP had a significant effect ($p \leq 0.01$) on flowering behavior across the five genotypes evaluated for each of the three variables studied (Table 2). The remaining contrast compares EP-L and EP-H averages and was also significant ($p \leq 0.01$) but only for D1F and HGT. Part of the variation due to the Photoperiod*Genotype interaction was also partitioned into five orthogonal contrasts comparing, for each genotype, the average of plants growing under DN and EP conditions. All contrasts, except for G2 (GM3893-65), were significant ($p \leq 0.01$), indicating a consistent response from the genotypes evaluated. In the case of GM3893-65 (the *asparagus* genotype) the contrast reached significance ($p \leq 0.05$) for D1F and NBL, but barely failed to reach significance for HGT.

Figure 2 presents the averages for the three light conditions (DN, EP-L, and EP-H) of each individual genotype and across genotypes. In general, there is a qualitative and consistent separation between DN and the EP averages, demonstrating the advantage of photoperiod extension. Differences between EP-L and EP-H averages were not consistent, particularly for NBL. In some cases, they were not statistically different, and in some cases plants growing under EP-H tended to have a better response (fewer days for D1F, lower HGT or higher NBL). Exceptionally (NBL in CM 4919-1), performance under EP-L was better than under EP-H. The advantages of extending the photoperiod, in the case of GM 3893-65 (*asparagus* cassava), were not consistent. There was a positive association between D1F and HGT when genotypes or light conditions were considered. As expected, there was a negative relationship between these two variables and NBL. Figure 3 provides a visual illustration of the effects that EP has on the five genotypes.

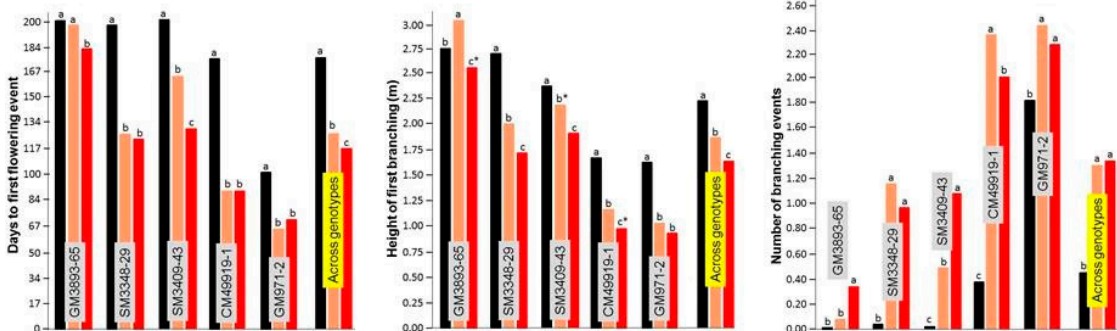

**Figure 2.** Averages for days to 1st flowering, height of 1st branching and number of branching events in five genotypes and across genotypes grown under normal photoperiod (black bars), and extended photoperiod based on low (pink bars) and high (red bars) light intensity. The comparison in individual genotypes and across genotypes with different letters, are statistically significant at *p* < 0.01 (except cases identified with an asterisk where differences were significant at *p* < 0.05). The statistical significances were derived from the analysis of variance presented in Table 2. * Differences were statistically significant at *p* < 0.05.

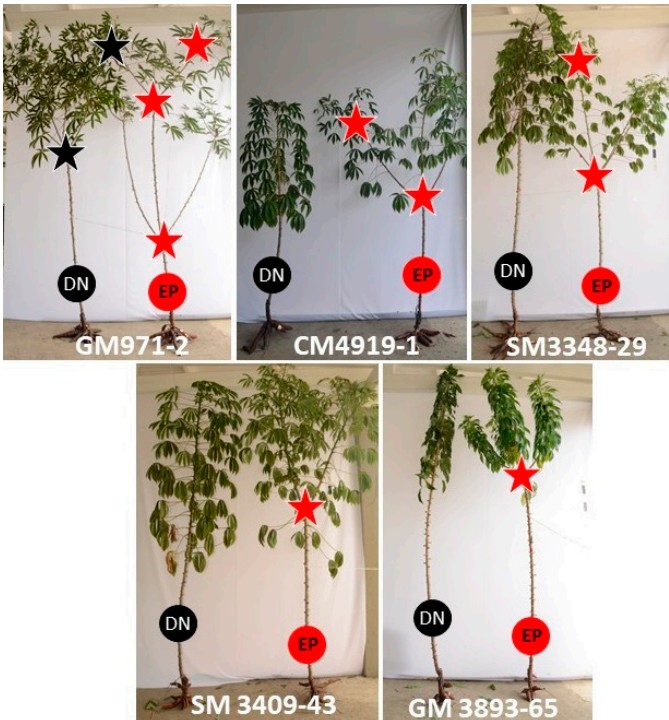

**Figure 3.** Comparison of plants grown under dark night (DN) and extended photoperiod (EP) for each of the five genotypes included in Experiment 1. For each genotype, plants were randomly selected from DN (left plant in each photograph) or EP plots (right plant in each photograph). Branching events are highlighted with black or red stars respectively for DN and EP.

Results for the second season of Experiment 1 (based on the use of 5LEDs only) validated the earlier finding. Rather than presenting ANOVA tables for the additional studies, Figure 4 presents a chronological description of flowering in the five experimental genotypes in three consecutive growing seasons (except SM 3409-43, which was not included in the 2018/19 trial). This figure combines the information from the two growing seasons of Experiment 1 (using individual LEDs) and the second growing seasons of Experiment 2 (based on the high mounted 50 W LED lamp with a reflector). Data from the two experiments were combined to demonstrate that plant responses were generally the same regardless of the light source and season. Contrasting performances of plants growing under EP

and DN can be easily visualized. In every genotype, flowering under EP occurred at a faster rate than under DN (the statistical significance of differences in the rate of flowering is presented in Table 3). Lines with the same color identify data from the same growing season. Solid and dotted lines depict performances under EP and DN, respectively.

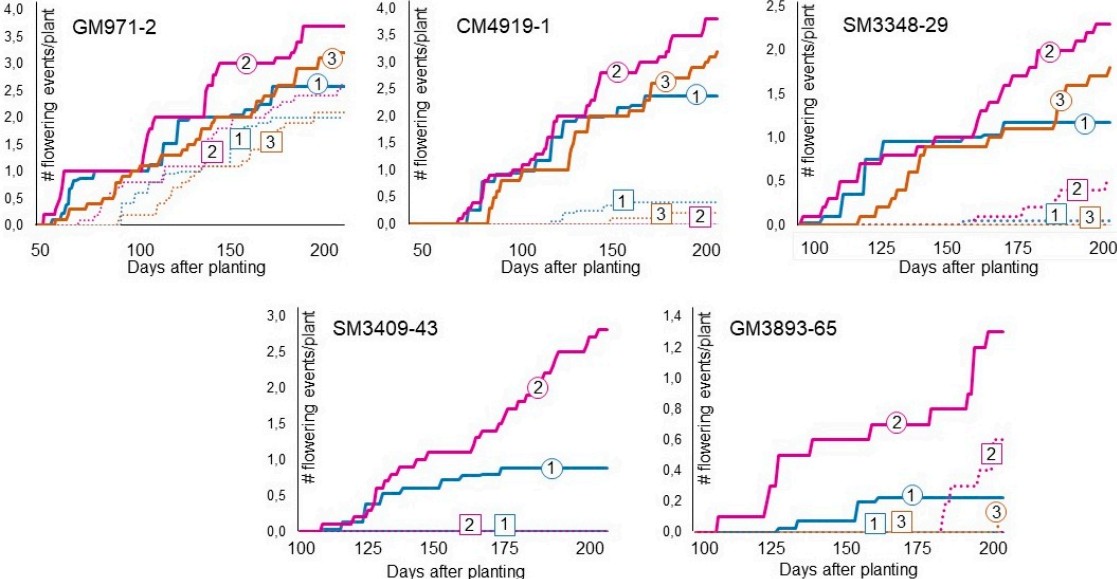

**Figure 4.** Average number of branching (e.g., flowering) events per plant in five genotypes evaluated in three consecutive harvesting seasons (except for genotype SM 3409-43). The horizontal axis presents the days after planting. Colors and numbers identify experiments and seasons: (1) Experiment 1 in 2016/17; (2) Experiment 1 in 2017/18; and (3) Experiment 2 in 2018/19. Numbers in circle and solid lines identify responses under extended photoperiod (averages of the four different light treatments for Experiment 1 in 2016/17). Numbers inside squares and doted lines identify the progress of flowering in the check plots under normal (dark) night conditions.

**Table 3.** Summary of relevant results from Experiments 1 and 2. **Top**: linear regression coefficients of the average number of flowering events. Within parenthesis are the respective standard errors for the coefficients. **Middle**: Average number of flowering events per plant 200 DAP. **Bottom**: average number of days to the second branching event. Within parenthesis, is the percentage of plants that flowered twice (or more) [a].

| Genotype | 2016/17 | | 2017/18 | | 2018/19 | |
|---|---|---|---|---|---|---|
| | DN | EP | DN | EP | DN | EP |
| **Regression Coefficients on Number of Flowering Events on DAP (Standard Error of Coefficient)** | | | | | | |
| GM 971-2 | 0.190 (0.014) | 0.521 (0.026) | 0.114 (0.006) | 0.176 (0.007) | 0.102 (0.004) | 0.151 (0.004) |
| CM 4919-1 | 0.037 (0.007) | 0.497 (0.029) | No flowering | 0.156 (0.008) | 0.009 (0.002) | 0.166 (0.005) |
| SM 3348-29 | 0.006 (0.000) | 0.215 (0.034) | 0.017 (0.003) | 0.085 (0.006) | No flowering | 0.098 (0.005) |
| GM 3893-65 | No flowering | 0.040 (0.009) | 0.017 (0.004) | 0.045 (0.005) | No flowering | 0.007 (0.002) |
| SM 3409-43 | No flowering | 0.162 (0.021) | No flowering | 0.097 (0.006) | N.A. | N.A. |
| **Average Number of Flowering Events (or Branching Levels) per Plant 200 Days after Planting [b,c]** | | | | | | |
| GM 971-2 | 2.00b | 2.58a ** | 2.30b | 3.70a ** | 2.10b | 3.10a ** |
| CM 4919-1 | 0.40d | 2.38ab ** | 0.00c | 3.50a ** | 0.20c | 2.90a ** |
| SM 3348-29 | 0.05d | 1.18c ** | 0.40c | 2.20b ** | 0.00c | 1.60b ** |
| GM 3893-65 | 0.00d | 0.23d* | 0.40c | 1.71b ** | 0.00c | 0.00c [NS] |
| SM 3409-43 | 0.00d | 0.88c ** | 0.00c | 2.50b ** | N.A. | N.A. |
| Across | 0.43 | 1.37 ** | 0.58 | 2.60 ** | 0.58 | 1.90 ** |

**Table 3.** *Cont.*

| Genotype | 2016/17 | | 2017/18 | | 2018/19 | |
|---|---|---|---|---|---|---|
| | DN | EP | DN | EP | DN | EP |
| Average Number of Days from Planting to 2nd Branching (% of Plants that Flowered) | | | | | | |
| GM 971-2 | 152.2 (95%) | 118.2 (100%) | 137.6 (80%) | 107.0 (100%) | 169.1 (90%) | 126.7 (100%) |
| CM 4919-1 | No flowering | 124.9 (100%) | No flowering | 119.3 (100%) | No flowering | 136.0 (100%) |
| SM 3348-29 | No flowering | 167.5 (21%) | No flowering | 169.7 (90%) | No flowering | 198.4 (100%) |
| GM 3893-65 | No flowering | No flowering | No flowering | 194.3 (60%) | No flowering | No flowering |
| SM 3409-43 | No flowering | 169.3 (10%) | No flowering | 168.1 (100%) | N.A. | N.A. |

[a] DN: dark nights; EP: extended photoperiod; DAP:days after planting; N.A.: Not available. [b] Averages for genotype (within growing season and photoperiod condition) followed by the same letter are not statistically significant at the 5% probability level (Tukey's Adjusted Multiple Comparisons test). [c] Contrasts between DN and EP for each genotype and across genotypes for each growing season: NS, non-significant; *, ** statistically significant at 5 and 1% probability level, respectively.

Differences were very noticeable except, perhaps, for GM 971-2, which was the only genotype with substantial flowering under DN. Data from this genotype (top left of Figure 4) provides a good illustration of the meaning of each curve. The solid purple line (# 2 within a circle) presents data from the second growing season (2017–2018) of Experiment 1. The average number of branching events began to rise soon after 50 DAP and by 60 DAP reached the value of 1.0. This means that 60 DAP all of the ten plants within the row had flowered for the first time. The average remained unchanged for 40 days, until the second flowering event began at about 104 DAP, and finished in all plants in this genotype ten days later. The third branching began 136 DAP and was completed by 145 DAP. The fourth branching event began by 171 DAP, but only seven of the ten plants flowered a 4th time. There was a clear stepwise progression of flowering, which was not as apparent in other genotypes or harvesting dates.

CM 4919-1 showed a clear-cut contrast between EP and DN in Figure 4. The EP treatment (three solid lines identified by circled numbers) flowered earlier and faster than DN (dotted lines identified with the squared numbers). It also appeared that flowering occurred earlier in the second season (2017/18) than in the first one (2016/17), while flowering in the third season (2018/19) was intermediate. Results from SM 3348-29 were very similar to those of CM 4919-1, but flowering began considerably later and, consequently, NBL was lower. SM 3409-43 was grown only within Experiment 1. It also showed a strong response to EP. Results from GM 971-2 are more difficult to describe because this is the only genotype that flowered consistently under DN. Still, in every case, plants growing under EP tended to flower earlier than those growing under DN in the same season (Figure 4). The last genotype (GM 3893-65), the so-called "*asparagus*" cassava, was very recalcitrant. Few plants from this genotype flowered under DN and only during the second season of Experiment 2. For the same experiment, all plants flowered at least once under EP. In the 2016/17 there was a marginal response to EP, which nonetheless, was better than for DN. GM 3893-65 did not flower at all in the 2018/19 season.

Table 3 summarizes the main information obtained from the three growing seasons in which experimental genotypes were illuminated all night long with individual LEDs or the 50 W lamps. Plots similar to those presented in Figure 4 were used for the regression analysis of NBL on time. The regression coefficients and their respective standard errors are listed on top of Table 3. In every case, the regression coefficient for EP was significantly higher than for DN (in each experiment and genotype combination). This provides statistical evidence for the differences depicted in Figure 4 and implies that EP accelerated the flowering process in every genotype.

The average NBL per plant 200 DAP is also presented in the middle of Table 3. This information is useful to compare flowering patterns of the different genotypes, as well as to assess the effects of extended photoperiod. GM 971-2 was the only genotype that flowered twice (or more) 200 DAP under DN. All other genotypes either failed to flower or did so only once and in few plants (averages between 0 and 1). EP always increased ($p \leq 0.01$) the number of flowering events 200 DAP compared with DN, except for GM 3893-65 in the first ($p \leq 0.05$) and the third (non-significant differences) growing season. Asterisks under the EP column for each growing season indicate the significance of the contrast

between EP and DN for each genotype (also across genotypes). It is clear that EP consistently increased the number of branching levels in the different genotypes compared with DN.

Table 3 also provides information on average number of days required for each genotype to flower a second time. Analysis of variance was not performed because many plots would have missing data (e.g., did not flower). It is usually in the second flowering event that breeders start making crosses and can obtain seeds in pollination nurseries. Only GM 971-2 flowered a second time under DN in every growing season. In EP, on the other hand, every genotype flowered at least twice (except for GM 3893-65 in the first and third experiments).

### 3.2. Assessing the Minimum Light Intensity to Induce Earlier Flowering

The first growing season of Experiment 2 (2017/18) served two purposes: (1) to test the effectiveness of the 50W lamp with reflector as a practical alternative to the cumbersome use of individual LEDs; and (2) to determine the minimum light intensity required for the induction of a significantly earlier flowering. Figure 5 presents the most relevant information from this evaluation. The number of times each plant flowered and branched (NBL) is presented in Figure 5A individually for each plant. The data presented corresponds to the status of plants by December 12 (148 DAP) when there was a good separation between fast and slow-flowering plants. Data from CM 4919-1 and GM 971-2 are in the top and bottom of Figure 5A, respectively. The yellow ring in Figure 5A marks the proposed limit of plants receiving enough stimulus from the LED lamp to elicit earlier flowering. In the case of CM 4919-1, the average number of branching events for plants growing outside this limit was much lower than those inside (0.6 vs. 2.4). In the case of GM 971-2, the averages of branching events outside and inside the limit were 1.9 and 2.4, respectively. The average number of branching events at 148 DAP in the control plots (DN) of GM 971-2 and CM 4919-1 were 1.8 and 0.0, respectively (data taken from the second season of Experiment 1, which was adjacent in the same field plot).

The relationship between NBL through the end of the evaluation (206 DAP) and the distance of each plant to the source of light is presented in Figure 5B. NBL were highest (4.0) in plants directly under the LED lamp, and decreased linearly with distance from the lamp with a regression coefficient of $-0.638 \pm 0.049$ (Figure 5B), which was significantly different from zero ($p < 0.0001$; $R^2$ of 0.77). The average NBL at 206 DAP in the control plots (DN) of GM 971-2 and CM 4919-1 were 2.7 and 0.0, respectively. Results from the initial evaluation of the 50 W LED lamp, therefore, agree with those presented in Table 3 showing that the extended photoperiod results in an induction of earlier flowering in the cassava genotypes evaluated.

Figure 5C shows the relationship of light intensity with distance from the reflector. The relationship agrees with the expected inverse square law (Intensity = $1/\text{distance}^2$). Given that full sunlight at noon provides about 2000 μmol of photosynthetically active radiation $m^{-2}$ $s^{-1}$, the light quantity $> 0.12$ μmol $m^{-2}$ $s^{-1}$ provided by the 50W lamp on plants immediately below the reflector is negligible for photosynthesis. Interestingly, plant responses tend to be asymmetrical, with better responses on the left compared with the right of Figure 5A. A lamp that is not perfectly horizontal may have resulted in this asymmetry. The reflector in the lamp is not perfectly square, but rather rectangular. The shape of the yellow ring, therefore, is slightly wider (6 m from the light) than tall (5 m from the light). It can be can be seen in Figure 5C that a 5–6 m distance corresponds to 0.02–0.03 μmol $m^{-2}$ $s^{-1}$. We suggest, therefore, that a minimum light quantity of around 0.02 μmol $m^{-2}$ $s^{-1}$ is needed to elicit earlier flowering. The yellow ring in Figure 5A marks this proposed threshold. This level of light intensity required to elicit earlier flowering is more detailed than the information from Experiment 1 (Table 1) and generally agrees with it.

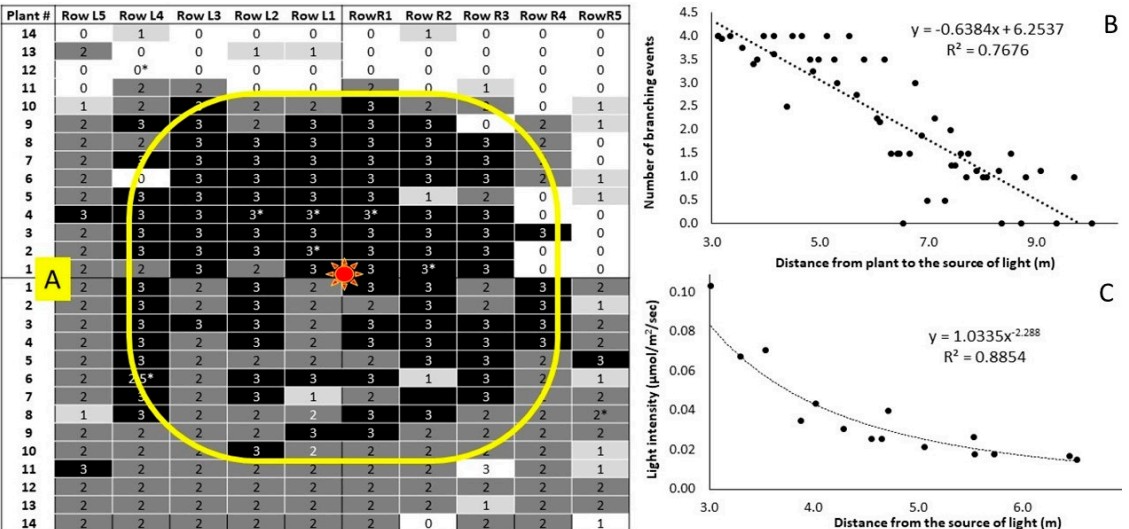

**Figure 5.** Results from the evaluation of the 50 W LED lamp treatment in Experiment 2 (**A**). The experimental area was planted in a $1.5 \times 0.5$ m grid with ten rows and 14 plants of CM 4919-1 (top) and GM 971-2 (bottom). The graphic indicates the number of branching events observed at 148 days after planting (DAP) for each plant. There were eight missing plants identified with an asterisk; the missing information for these plants was replaced by the average of the surrounding plants. Plants inside the yellow ring received enough stimulus to flower earlier. The red star marker indicates the position of the source of light. (**B**) relationship between average number of branching events 206 DAP and average distance to the source of light (both genotypes shown; data from plants at the same distance from the reflector were averaged). (**C**) Average intensity of illumination in relation to the distance from the lamp based on 53 measurements. The yellow ring in (**A**) indicates plants that had received >0.02 $\mu$mol m$^{-2}$ s$^{-1}$, the proposed threshold to induce earlier flowering (about 5–6 m from ground level to the center of the plot immediately below the lamp).

### 3.3. The Effect of Night Breaks (Experiment 3)

Figure 6 summarizes the response of flowering events in the two genotypes to the various night break (NB) treatments. In the case of GM 971-2 (left panel), there were no clear-cut differences between various NB treatments, though the check treatment (DN) for this genotype was generally the lowest throughout the duration of the experiment. In CM 4919-1 (right panel), no plant flowered under DN, while all the NB treatments had between 1.0 and 3.7 flowering events during the period of observation. In CM 4919-1, the NB60-10LEDs treatment elicited just one flowering event, while every other NB treatment induced at least 2.3 flowering events by the end of the experiment.

Regression analyses of NBL on time, similar to those presented at the top of Table 3, were conducted (data not presented to avoid duplication of the information provided in Figure 6). The lowest regression coefficients, among NB for GM 971-2, were those plots illuminated with 10LEDs (NB60, NB120, and NB180). They were, however, significantly higher ($p \leq 0.05$) than that for the DN treatment. Interestingly, the shortest night break (NB30-30LEDs) resulted in the highest regression coefficient for GM 971-2. Slightly smaller coefficients (in comparison with NB30) were observed for the two remaining breaks (90 and 240 min), both with the 60LEDs. Similar responses were observed for CM 4919-1. The regression coefficient for NB60-10LEDs was less than half compared with the remaining NBs, but significantly higher than that for the DN check. The second lowest coefficient was observed in NB120-10LEDs. The low light intensity common to these night breaks had a clear impact on the plant responses. Three NBs had similar regression coefficients (NB30-30LEDs, NB180-10LEDs, and NB240-60LEDs). The regression coefficient for CM 4919-1 under the NB90-60LEDs was significantly higher than any other coefficient for that genotype. The number of branching events 200 DAP in both genotypes was significantly higher ($p < 0.01$) under NB than DN (data not presented).

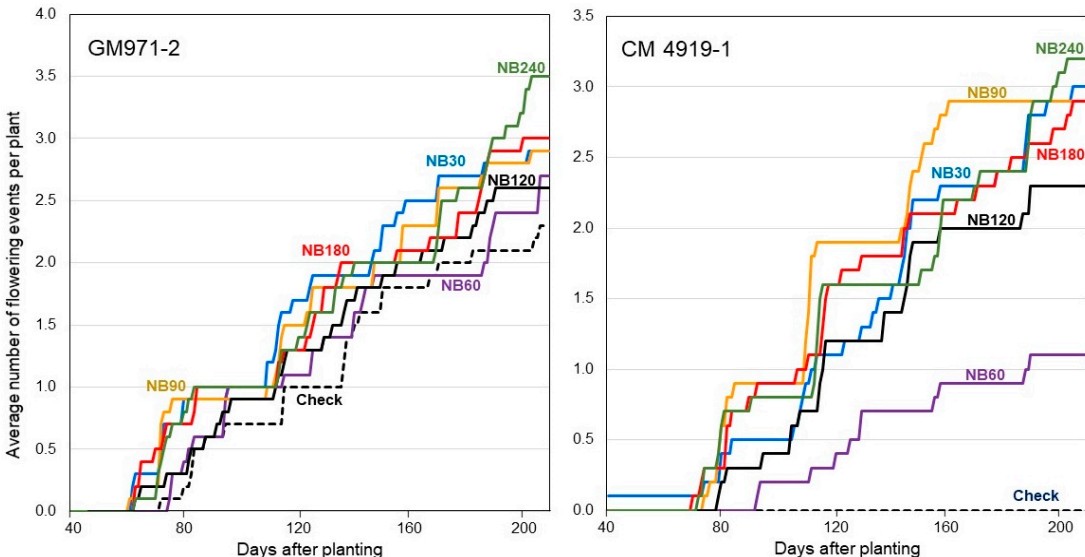

**Figure 6.** Effect of night breaks on the flowering patterns of two cassava genotypes. Results are based on ten plants per genotype and night break (NB) treatments were applied for various durations, centered on midnight. An intermediate light intensity was used for NB30. NB60, NB120, and NB180 were based on a low-intensity illumination. NB90 and NB240 were illuminated with a high intensity red-light source. See Methods and Materials for details.

Results from Experiment 3 indicated that NBs are indeed useful to elicit flowering from plants, but also that light intensity matters when plants are not illuminated all night long. A threshold of stimulus needs to be surpassed to elicit plant responses, this threshold can be reached through long NBs and low light intensity or short NBs relying on high light intensity.

### 3.4. The Impact of Photoperiod Extension in a Wide Range of Genotypes (Experiment 4)

A crossing nursery with 116 elite genotypes was grown in the 2018/2019 growing season under EP and DN conditions. The light sources for EP were 50W LED lamps with reflectors (Figure 1E) positioned on a 4.5 m square grid which guaranteed a light quantity > 0.02 µmol/m$^2$/s. Figure 7 illustrates the relationship between NBL and HGT of the first branching event (an indicator of earliness), under DN and EP. It is clear that genotypes that flower early (e.g., short HGT) tend to have more flowering events (NBL) during the growing cycle (between four and five times) than late-flowering ones. This trend was not changed under EP. Interestingly, average NBL was not increased under EP (2.67 and 2.61 in DN and EP, respectively). The most noticeable effect of EP was a reduction of the average HGT. More relevant is that under EP (right plot in Figure 7) the lowest NBL was ≥1.0. In other words, every genotype had flowered at least once.

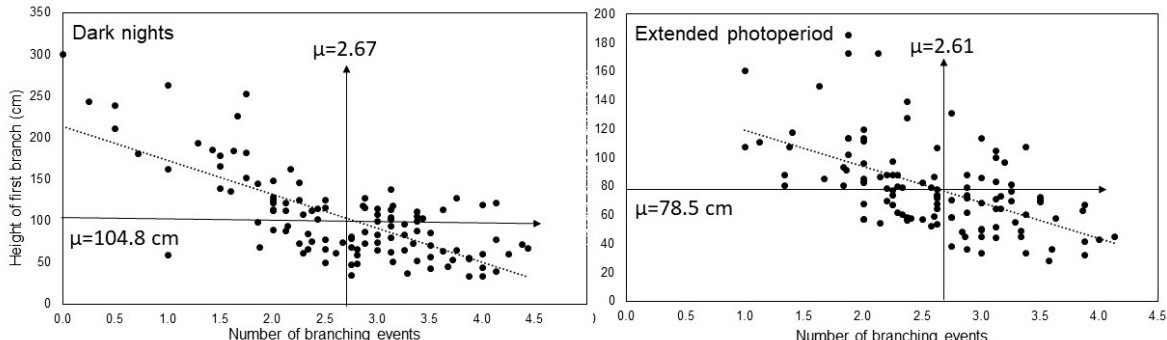

**Figure 7.** Relationship between number of branching events and height of the first branching in 116 genotypes grown under DN (**left**) and EP (**right**). The average values (µ) are shown for each treatment.

Based on our experience from previous experiments, it was expected that EP would result in an overall increase in NBL across the entire nursery. However, that was not the case, even though non-flowering genotypes had vanished under EP (every genotype had at least flowered once as shown in the right plot in Figure 6). This means that, in some genotypes, EP must have had a negative impact.

Figure 8 presents, basically, the same data as in Figure 7 except that the vertical axis depicts the difference between average number of branching events for each genotype under EP and DN, which is a measure of the extent to which EP hastened flowering. The horizontal axis ranks genotypes by the number of branching events under DN (horizontal axis in Figures 7 and 8 is the same). The information in Figure 8 suggests that the main effect of EP is to increase the number of flowering events (values > 0 in the vertical axis) in erect genotypes that normally do not flower or do so scarcely (left values in the horizontal axis). Figure 8 also shows that genotypes that normally flower profusely in the right of the plot ("*bushy*" types) had fewer branching events under EP (values < 0 in the vertical axis).

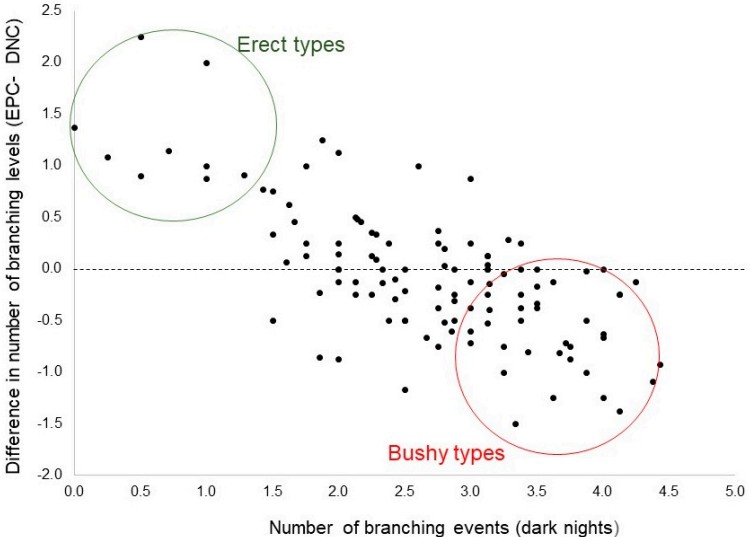

**Figure 8.** Extent to which extended photoperiod hastened flowering (difference between average number of branching events per plant under EP and DN as a function of average number of branching events per plant under DN in 116 elite cassava genotypes).

## 4. Discussion

The ultimate objective of this study was to overcome a critical problem that cassava breeders face worldwide: erect genotypes are the most desirable from the agronomic point of view, but almost impossible to use as progenitors for breeding purposes. The most important result in this study is that EP consistently induced earlier flowering in erect genotypes. Different light fixtures, light quantity, hours of exposure, growing seasons and genotypes consistently support this conclusion. The varying conditions used to compare DN and EP strengthens our confidence in the results and provides a broader scope for inferences. Moreover, this research provides a set up that can be used by breeders (even when electricity is not available all night long) to elicit earlier flowering in their crossing nurseries. The differential response of erect and bushy types to EP was unexpected. This may be due to varying genetic sensibilities to photoperiod similar to those observed between temperate and tropical maize [47]. Cassava breeding populations lack structure [48], partially, because genotypes from distant and diverse origin are often crossed thus erasing any distinctive feature they may have had. Predicting responses to EP based on the available information from the origin of the 116 progenitors included in Experiment 4 was not possible. Future efforts will focus on genotyping large sample of genotypes (167 progenitors for seven breeding pipelines) in search of molecular markers (SNPs) that can efficiently predict the response to EP.

It was occasionally observed, within the 10-plant row, that one or two plants had a very different flowering pattern. For example, nine plants would flower (or branch) three times during the duration of the trial but the remaining plant in the plot would not do so, not even once. Similarly, in a given plot all plants except one would fail to flower. Outlier plants, however, did not prevent identifying clear responses from the different genotypes to the light treatments. Epigenetic effects or other unknown factors may be the source of this variation within plots, including the possibility that stem cuttings used for propagation were obtained from mother plants that had been exposed to longer or shorter photoperiods (there is preliminary information supporting this hypothesis). It has to be recognized, however, that genotype-by-environment interactions play an important role in the impact of extended photoperiod. Efficient sexual reproduction is not essential for cassava [45]. Flowering, therefore, does not need to be fine-tuned, which could explain some of the unaccounted for variation reported in this study and empirical observations from crossing nurseries.

In general, conditions from the second growing season (2017/18) tended to promote earlier flowering than those from the 2016/17 and 2018/19 seasons (Figure 4). This suggests year-to-year environmental effects on flowering patterns for the genotypes evaluated. A feasible explanation for this variation is night temperatures. Average minimum temperatures through 60 DAP in 2016, 2017 and 2018 were respectively 19.6, 18.6, and 19.0 °C. The numbers of days with minimum temperature $\leq 18$ °C were 5, 15, and 10, respectively. It has been reported in a number of species, including cassava, that warm temperatures delay flowering [42,49]. It is possible, therefore, that low temperatures at night during the early flower induction period (first two MAP) may enhance the impact of EP. Studies conducted under controlled conditions at Cornell University had also found a synergistic effect of low temperatures with extended photoperiod for the induction of earlier flowering [50]. Low temperatures at night may also explain why extended photoperiod induced earlier flowering in Palmira (almost 1000 m.a.s.l.), but not at low altitude tropical locations (unpublished results).

Results from Experiment 3 were useful to demonstrate not only the efficiency of night breaks but also to highlight the complex interactions between duration of the break and the quantity of light provided by different sources. Night breaks would be justified when extended photoperiod is achieved using solar panels where no other source of electricity is available. Stored energy can then be released in a short period of illumination with many LEDs or in longer breaks using fewer LEDs.

The average number of flowering events at 200 DAP, (Table 3) is useful information considering that, generally speaking, the first flowering inductive event results in a sterile inflorescence in cassava. Pollinations in crossing nurseries, therefore, generally start at the second branching event. EP always increased the number of flowering events compared with DN (Table 3, Figure 4). However, the practical implications were different depending on the genotype. Since GM 971-2 flowered at least two times in every experiment under DN, the advantage of EP for this genotype is marginal: at best seed would be produced only a few weeks earlier. On the other hand, EP offers a major advantage for CM 4919-1. This is an outstanding genotype released in Colombia as CORPOICA-Verónica. Its erect plant architecture and corresponding excessively late flowering has prevented the extensive exploitation of this genotype as progenitor [51]. Table 3 and Figure 4 show that CM 4919-1 failed to flower more than once under DN, and only a few plants had done so by 200 DAP (averages < 1). In contrast, under EP, the average NBL for CM 4919-1 was always higher than two. Moreover, in the 2017/18 season plants from this genotype had flowered more than three times (Table 3, Figure 4). In practical terms, this means that EP would allow seed production for this valuable genotype within a year after planting. For the remaining genotypes, EP always resulted in a higher NBL, regardless the source of light used. However, in several cases, more than 200 days would have been required for plants to flower a second time.

The fertility of female flowers produced under EP seems to be no different from those under DN conditions. There was a normal production of seeds in the open-pollinated flowers. Moreover, the germination capacity of botanical seed produced (after the combination of EP and pruning young branches) was 85%, without any abnormal appearance in the resulting seedlings [45]. EP would allow

shortening the breeding cycle of erect progenitors (the most desirable ones) by a year. It would also facilitate the introduction of inbreeding, through successive self-pollinations. Previous inbreeding attempts failed because it favored early branching phenotypes [2]. EP can be used in combination of pruning at the first flowering event (which prevents abortion of inflorescences) to further accelerate and increase seed production [45].

## 5. Conclusions

Results from this work clearly indicate that EP induces earlier flowering in late-flowering cassava genotypes. Since earlier flowering under EP was particularly noticeable for genotypes that assume an erect plant architecture under DN, there is a clear evidence that responses are genotype-dependent. Earlier flowering was observed in plants exposed to $> 0.02$ μmol of photons m$^{-2}$ s$^{-1}$ all night long. However, there seems to be a quantitative change in this proposed threshold if night breaks are used. The effectiveness of NBs allows the use of solar panels in areas where reliable access to electricity is not available.

These experiments were conducted in Palmira, Colombia (at 1000 m above sea level). Thus, this location is characterized by relatively cooler nights compared with tropical locations at sea level. It has been reported that the effect of photoperiod extension often interacts with temperature. Further studies need to be carried out at other locations to assess the potential influence of night temperature on the relationship between flowering and photoperiod. However, EP had positive impacts on germplasm from Uganda and Thailand when tested at 1000 m.a.s.l (Robert Kawuki and Pasajee Kongsila, personal communication).

**Author Contributions:** Conceptualization, T.L.S. and H.C.; methodology M.P., N.M., S.S. and P.T.H.; formal analysis, H.C., M.P. and J.C.; investigation, M.P., N.M. and S.S.; resources, H.C.; data curation, M.P.; writing—original draft preparation, T.L.S. and H.C.; visualization, M.P. and H.C.; supervision, H.C.; project administration, H.C.; funding acquisition, H.C. All authors have read and agreed to the published version of the manuscript.

**Funding:** This research was funded by Bill & Melinda Gates Foundation and the Department for International Development of the United Kingdom through the Next Generation Cassava Breeding Project, Agreement No. 67724-10566 (under prime agreement No. OPP1048542).

**Acknowledgments:** We would like to thank Juan de la Cruz Jimenez and Frank Montenegro for their support in measuring light intensities and the valuable and reliable field support by Eugenio Bolaños. Anonymous reviewers made valuable comments, corrections and suggestions. The authors of this manuscript would like to thank them for their valuable contributions.

**Conflicts of Interest:** The authors declare no conflict of interest.

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
