# Peer review of "Induction of Earlier Flowering in Cassava through Extended Photoperiod"

_agronomy, doi:10.3390/agronomy10091273_

Round 1

Reviewer 1 Report

Here the authors very convincingly show that placing a LED lamp in a cassava field site can be used to induce flowering. A series of multi-year field trials with artificial light was used to reveals that proximity to after-dusk lighting leads to promotion of flowering induction. Figure 5 is breathtaking!

Major thoughts:

The first sentence of the introduction does not at all encapsulate the problem being tackled. Consider the problem and rewrite the abstract. This is all about getting cassava to flower (presumably to make crosses).

line 145: explain exactly who manufactured the LEDs and exactly how they were assembled.

It is useful in the methods to have 4 or 5 words that Columbia has virtually no seasonal changes in photoperiod nor seasonal weather changes (the later are often seen at other equatorial zones).

Figure 2 lacks a statistical presentation and thus cannot be assessed.

For the future: can the anthers and stigmas of these flowering plants be useful in controlled breeding schemes? I would like 2 sentences on that narrative, please. This could be in the CONCLUSION from line 589. How would breeding actually happen in these plants and what if the flowers lack fertility in either gender? Very small thoughts, max two sentences on these thoughts.

Minor thoughts:

Do not use the term “flux”. Consider photons per meter squared per second, which is E.

line 145: Each source of light contributed negligible photon 146 flux density for photosynthesis. This needs clarification or should be deleted. At a basic level, it really does not matter why the rings and lamps induce flowering. It could be good for the discussion to reflect, but it is not relevant per se.

line 152: Lights were turned on at sunset and turned off at sunrise.: scientifically this needs to be a discussion of course before and after dawn, called zeitgeber time. Dawn is ZT = 0. And as Columbia has no photoperiod changes in seasons, then dusk is essentially always ZT = 12. Use these scientific terms please.

line 153: therefore, lasted all night long; that is lay-person jargon. Please be specific. X-hours of light were added after ZT = Y.

There are numerous typographical and punctuation errors in the manuscript. These do not detract from the core message, but the paper would look cleaner if these was tidied up.

Author Response

We truly appreciate the effort and valuable corrections and suggestions made by this reviewer. Our response to the review can be found in the attached document.

Reviewer 2 Report

The target of this research is to develop a practical method to speed up breeding through enhancing flowering process. The focus on the use of red light for the whole nights with different light intensities or only the use of night break. The success of this work will speed up breeding by shortening the breeding cycle by a year and this is why this work is valuable. Although the introduction is nicely written, the discussion part was shallow and did not discuss all the experiments which were conducted.

The points to be worked out:

  • Lines 30 till 37 should be deleted (instructions)
  • The author tends to mix between clones and genotypes. Clones are not synonymous word to genotype. In this study 5 genotypes and 116 genotypes were use (they are genetically different). This should be corrected in throughout the text.
  • Line 42: “)” should be added
  • Line 135: ‘SM 3409-4’ is mentioned however ‘SM 3409-43’ was used overall the article. This should be consistent.
  • Line140: 116 genotypes not clone, same for Line 185, Line 339, Line 503, Line 515….
  • Line 240: check the grammar
  • Experiment 1: lots of effort were placed in running the 4 different light intensities, replicates per treatment, and data collection. However, at the moment of data analysis, the data was combined: 5 and 10 LED together, and 30 and 60LED together (to simplify the analysis). This is essential part of the results as it was used to make decision for season 2. There was significant difference (Table 2) EP-L vs. EP-H, Which treatment was giving better results (based on Figure 2, EP-H was better)? The reason of using 5LED for season 2 was not clear. Are there significant differences between 5 and 10LED treatments?
  • Also, are there differences among the 5 genotypes in responding to different light intensities? The author could mention (for example) only the ones which was significant to keep it simple.
  • Table 2: G2 (GM3893-65) did not show strong effect of EP. This genotype does not flower under normal light condition and with EP showed positive response however that was not clear if it responded to low or to high intensities (line 302). And based on the answer, how this genotype is expected to respond to Experiment 2 (season 2) if light intensities are compared between Experiment 1 (high light intensity) and Experiment 2?
  • Figure2: where is key colour?
  • Figure3: line 333, should be five not six genotypes.
  • Line 385: GM3500-2 is mentioned however it is not one of the 5 studied genotypes!!
  • Figure4: what data were used for Experiment 1 season 1 for the comparison with season 2? In season 1, 4 different treatments were used but only one in season 2? Did the author use the averages of all treatments of season 1? This needs to be clarified.
  • In Experiment 1, season 2 overall performed significantly better that season 1 considering using only 5LED. Is this difference due to light intensity? Also why these experiments were not planted on the same/close time:
  • Experiment 1, season 1: planted on 8 august 2016, and season 2: planted on 2nd June 2017
  • Experiment 2: season 1: planted on 18th July 2017, season 2: planted on 6th June 2018.

There are two months difference in experiment 1? Why is this? Could this explain the difference in results of season 1 and 2? The author suggests the night temperature as an influencing factor, however the difference of one degree (19.6 and 18.6 C) could be of limit influence!!

  • Line 471: CM4919-1 did not flower under DN in experiment 3. This is contradicting with experiment 1 results!! What is the explanation for this?
  • Figure6: NB60 treatment gave different results compared with other treatments which have higher and lower intensities/durations…any explanation? We do not see clear gradation in response to different treatments of Night breaks neither clear cut between different treatments. This needs further discussion.
  • It was interesting to see that the erect types responded differently to light treatment compared with Bushy types. Does this refer to different mechanisms of controlling flowering process in these two types? Do they have different origin?…the author did not elaborate in the discussion on this point!
  • The author did not elaborate in the discussion part about the use of night breaks and future us. Would they be as effective and replace the whole EP treatments? Any plan for validation of this results?
  • Line 434: this results of Experiment 1 or 2?
  • Experiment 2 season one, 2 genotypes (GM971-2, and CM4919-1) and in season 2 four genotypes (GM971-2, CM4919-1, GM3893-65, SM 3348-29) were tested. In data analysis sessions we could not find the analysis/comparison of the two seasons’ result considering that 2 genotypes were tested in two seasons? In figure 4 only the results of season 2 were used. Why? How the author dealt with the two years results of GM971-2, and CM4919-1?

Author Response

We appreciate the effort made by this reviewer and the valuable corrections and suggestions made. Our response to the review can be found in the attached document.
